# What Are the Key Workplace Influences on Pathways of Work Ability? A Six-Year Follow Up

**DOI:** 10.3390/ijerph16132363

**Published:** 2019-07-03

**Authors:** Jodi Oakman, Subas Neupane, K.C. Prakash, Clas-Håkan Nygård

**Affiliations:** 1Centre for Ergonomics and Human Factors, School of Psychology and Public Health, La Trobe University, Melbourne, VIC 3086, Australia; 2Unit of Health Sciences, Faculty of Social Science, Tampere University, 33014 Tampere, Finland

**Keywords:** work ability, work environment, physical hazards, psychosocial hazards, multisite pain, musculoskeletal pain, trajectories

## Abstract

Objective: To study the trajectories of work ability and investigate the impact of multisite pain and working conditions on pathways of work ability over a six-year period. Methods: The longitudinal study was conducted with Finnish food industry workers (*n* = 866) with data collected every 2 years from 2003–2009. Questions covered musculoskeletal pain, physical and psychosocial working conditions (physical strain, repetitive movements, awkward postures; mental strain, team support, leadership, possibility to influence) and work ability. Latent class growth analysis and logistic regression were used to analyse the impact of multisite pain and working conditions on work ability trajectories (pathways). Results: Three trajectories of work ability emerged: decreasing (5%), increasing (5%), and good (90%). In the former two trajectories, the mean score of work ability changed from good to poor and poor to good during follow-up, while in the latter, individuals maintained good work ability during the follow-up. In the multivariable adjusted model, number of pain sites was significantly associated with higher odds of belonging to the trajectory of poor work ability (Odds ratio (OR) 4 pain sites 2.96, 1.25–7.03). Conclusions: A substantial number of employees maintained good work ability across the follow up. However, for employees with poor work ability, multisite musculoskeletal pain has an important influence, with effective prevention strategies required to reduce its prevalence.

## 1. Introduction

An ageing population means longer working lives are needed to support labour supply and to provide an adequate income in retirement [1,2,3]. Maintenance of good work ability, which includes physical and mental capacities, across the life course is important to enable employees to sustain an extended working life [4]. Poor health and work ability are key determinants of early exit from work [5,6]; hence, identifying potentially modifiable workplace factors to address these issues should be included as part of an overall strategy to extend working lives. To contribute to achieving this goal, examination of work ability pathways over time is required to identify key workplace factors which influence an individual’s work ability.

Dimensions of work ability comprise both individual factors (health and functional capacity, skills and knowledge required to complete the work and attitudes and motivations towards work) and work and work-context factors (supervisory support and physical, psychosocial and organizational work-related factors) [7]. The impact of having low work ability is significant; a 28-year follow up found that poor work ability at midlife was linked with higher odds of morbidity and disability during retirement and in old age. Poor physical and psychosocial working conditions have been associated with declining work ability [8,9].

Pathways of work ability have been examined previously [10,11]. However, some limitations apply: Feldt et al [10] only covered managers in their study whilst other studies have focused specifically populations of older workers [11]. A previous study on the same population reported here also examined work ability; however, the current study utilises a longer follow up period than previously where Neupane [12] reported over a four-year follow up that multisite pain (MSP) was a strong predictor of work ability. Work ability is assessed against an individual’s lifetime work ability and so is best suited to longitudinal analysis over an extended time period. Tuomi and colleagues [13] reported on work ability over an 11-year follow up and found that role ambiguity and physical work strongly were associated with decreased work ability for both males and females. Importantly, they also found over the long follow up period the relative influence of variables changed, which suggests the need for an extended follow up period to analyse the impacts of working conditions on work ability.

The relationship between multisite musculoskeletal pain (MSP) and work ability has been previously reported [12,14] with MSP having a higher prevalence compared to single-site pain [15,16,17] and is associated with a range of adverse outcomes including: poor work ability [12,18], long term sickness absence [19], and early retirement [20,21].

Improved understanding of the influence of working conditions on the development of pathways of work ability over time will enable more focused interventions to be implemented in the workplaces.

### Purpose of Study

This study aimed to examine the pathways (trajectories) of work ability over 6 years of follow-up. The second aim is to explore whether the baseline psychosocial or physical working conditions and multisite musculoskeletal pain influence work ability pathways.

## 2. Materials and Methods

### 2.1. Study Design and Data Collection

Data for the study were collected from employees from a large Finnish Food Industry Company via surveys over a six-year period. Blue and white collar employees were involved; the former engaged in more physically orientated work and the latter in administrative and managerial roles [14]. Surveys were completed anonymously. Questionnaires were distributed at the work place but were not addressed to individual employees, so personal reminders could not be sent. Respondents could respond anonymously or provide their name and consent for linking survey data with register data obtained from the company personnel registers [22].

In 2003, a 63 percent (*N* = 873) response rate was obtained. In 2005, 2007 and 2009, 1201, 1400 and 1398 people replied to the questionnaire, respectively. For inclusion in the current analysis, participants must have responded to the baseline survey and at least one of the follow-up surveys. A total of 866 people responded to the work ability question at baseline and first follow-up survey; 542 people in the baseline and second round of follow-up and 417 people replied to the baseline and last follow-up. Respondents who responded to the baseline survey were aged between 18 and 64 years (mean age 40.5 ± 11.1); almost 70% were women and blue-collar workers.

Ethics approval for the study was provided by the Pirkanmaa Hospital District (approval number R03043), Tampere, Finland.

### 2.2. Measurement of Variables

#### 2.2.1. Work Ability

Work ability was measured in all four surveys with the question “how is your current work ability compared with life time best?”, with responses from 0 (absolutely incapable of work) to 10 (work ability at its best). The use of a single item has been confirmed as an acceptable measure of work ability [23,24]. A continuous score of work ability was used to model the trajectories.

#### 2.2.2. Musculoskeletal Pain

Musculoskeletal pain at baseline was assessed with a modified version of the validated Nordic Musculoskeletal Questionnaire [25]. Questions on perceived pain, ache or numbness in four anatomical areas (hands or upper extremities; neck or shoulders; lower back; and feet or lower extremities) during the preceding week from 0 (not at all) to 10 (very much) were asked. The variables were dichotomised at the median score (less than or equal to median: 0 = mild; more than median: 1 = severe). The cut-off values for upper extremities, neck and shoulder, lower back and lower extremities were 4, 5, 2 and 2, respectively. The dichotomised variables were summed into a variable, expressing the number of areas with severe pain (from 0–4) [14,22].

#### 2.2.3. Physical Working Conditions

Physical strain at baseline was measured as the rating of perceived exertion (RPE) with the question “How physically hard/exhausting do you feel your job is on a normal work day?” on a scale from 6 (not at all) to 20 (very much) [26]. The physical strain was dichotomised using a median value as the cut off point (6–13 as low and 14–20 as high physical strain).

Other variables related to physical working conditions were assessed at baseline through questions on ‘repetitive movements’ and ‘awkward postures’. A scale of 1 (not at all) to, 5 (very much) was used and dichotomized into ‘Low’ and ‘High’ at the median value (cut-off value 3 for both).

#### 2.2.4. Psychosocial Working Conditions

Psychosocial factors from baseline are used in this study, and have been described in detail elsewhere [27], in the following areas: ‘incentive and participative leadership’, ‘team support’ and ‘possibilities to exert influence at work’ were asked with a response scale from 1 (totally disagree/very probably not) to 5 (totally agree/very probably) [28]. Responses were summed and divided by the number of variables used in the index. Cronbach’s αs of the measures were 0.71, 0.79 and 0.82, respectively. All psychosocial factors used in the analysis were dichotomised using the median value as the cut-off point, median or less as ‘poor’ and higher than median as ‘good’. Median values were 3.16, 3.16 and 3.20 for ‘incentive and participative leadership’, ‘team support’ and ‘possibilities to exert influence at work’, respectively.

Perceived mental strain at baseline was assessed using a modified version of the occupational stress questionnaire [29]: (“Stress means a situation in which a person feels excited, apprehensive/concerned, nervous or distressed or she/he cannot sleep because of the things on her/his mind. Do you feel this kind of stress nowadays?”) with a scale from 0 (not at all) to 10 (very much). The variable was dichotomised as “low” (0–4) and “high” (5–10) using the median value as the cut-off point using the median value 4 as the cut-off point.

#### 2.2.5. Other Covariates

Baseline information on age was categorized into two groups (<45 years, ≥45 years), and gender (male, female) and occupational class (blue-collar, white-collar) were used as other covariates.

### 2.3. Statistical Analysis

Latent class growth analysis (LCGA) was used to identify the developmental path (trajectories) of work ability. The linear function best fitted the patterns of change in the data using work ability as a continuous variable. Latent class growth analysis enables the identification of different developmental patterns over several measurement points. It is a special case of the growth mixture model given the assumption of homogeneity of growth parameters within a latent subgroup [30]. Individuals were included in the final analysis if they had responded to the baseline survey and at least one of the follow-up surveys. However, preliminary analysis was undertaken of those who responded to all four waves (*n* = 327) and the trajectory shapes were unchanged. Therefore, a decision was made to include all respondents who replied to the baseline (*N* = 866) and at least one of the follow-up surveys. The trajectory groups are illustrated by plotting mean levels of MSP against year of the survey (Figure 1).

The final model was chosen based on a range of fit criteria (see Appendix A), which include Akaike Information Criterion (AIC), Bayesian Information Criterion (BIC), sample size-adjusted BIC, entropy and proportion of trajectory group. In the fit criteria, a lower BIC, AIC and sample size adjusted BIC value and entropy close to one indicate a better model fit. Moreover, interpretability of the model was considered. Based on the above fit criteria, a three-trajectory model was determined as the most appropriate.

Baseline characteristics of subjects were examined by trajectory group using the Chi-Square test. Two of the trajectories (decreasing and increasing) were collapsed for analysis here, due to the similar characteristics in the representation of patterns, and called the poor work ability trajectory group to ensure enough statistical power in the regression models. The association between trajectories of work ability and baseline multisite pain adjusted for physical and psychosocial working conditions as well as socio-demographic factors were examined using binary logistic regression. Odds ratios (ORs) and their 95% confidence intervals (CIs) were used as the measure of associations. Models were built in four steps; the crude model, a second model was adjusted for covariates (age, gender, and occupational status) and physical working conditions (physical strain, repetitive movements, and awkward posture). The third model was adjusted for covariates and psychosocial working conditions (mental strain, leadership, team support, and possibility to influence). The final model was adjusted for all variables used in the previous models. The two-way interaction of each of socio-demographic variables, physical and psychosocial working conditions with number of pain sites with respect to poor work ability was tested. Only the significant interaction terms (team support and number of pain sites; possibility to influence and number of pain sites; occupational class and number of pain sites) are presented as a probability plot in the Appendix A. LCGA was analysed in Mplus v7.2 (Muthén & Muthén, Los Angeles, CA) and the regression analysis was performed in Stata 14 (StataCorp. 2015. Stata Statistical Software: Release 14. College Station, TX: StataCorp LP).

## 3. Results

The result of the final trajectory solution is presented in Figure 1. Three trajectories of work ability were identified: decreasing (*n* = 41, 5%), increasing (*n* = 40, 5%) and good (*n* = 786, 90%). The decreasing trajectory group comprised individuals with good work ability at the baseline, with a mean work ability score of 8.5, which then decreased during the follow-up to a mean of 4 or poor work ability at the final round of follow up. Similarly, individuals in the increasing trajectory group started with poor work ability at the baseline (mean score about 4.5), which then increased over the follow up period. The majority of the individuals maintained good work ability throughout the follow-up, with a mean work ability score of almost nine at the baseline, and in the last round of follow-up, there was a slight decrease to a mean of 8.3.

The levels of baseline socio-demographic and work-related characteristics of the studied population were significantly different for the three work ability pathways with the exception of age, gender, mental strain and leadership (Table 1). Individuals in the good trajectory group were more often white-collar employees, with less exposure to physically orientated work, had good psychosocial working conditions and to report either none or 1–2-site pain. In contrast, individuals in the increasing or decreasing work ability trajectory group were more often blue-collar employees, engaged in physically demanding work, and likely to report poor psychosocial working conditions and pain in three to four sites.

The association of the poor work ability trajectory with the number of pain sites at the baseline, working conditions and socio-demographic factors are presented in Table 2. In the crude model (Model I), poor work ability was strongly associated with multisite pain with higher odds replicating a dose response association, compared to the individuals with no pain. The associations remained statistically significant in the fully adjusted model (Model IV) when the model was adjusted for physical and psychosocial working conditions, age, gender and occupational class, and still maintained the dose-repose manner (OR for 3-site pain 2.45, 95% CI 1.00–6.00 and 4-site pain 2.96, 1.25–7.03).

White-collar employees had significantly lower odds of belonging to the poor work ability trajectory in the crude model, but the association no longer remained significant in the final model. Similarly, among working conditions, individuals with high physical strain, high repetitive movements, high awkward posture, poor team support and poor possibility to influence had higher odds of belonging to the poor work ability trajectory in the crude model. However, significant associations were lost when the models were adjusted as outlined in Model II, Model II and fully adjusted Model IV.

Interaction effects of team support and the number of pain sites, possibility to influence and number of pain sites and occupational class and number of pain sites with respect to poor work ability was estimated as a post-estimation effect (S1). Wider differences between good and poor team support and between good and poor possibility to influence were found, especially among those with three pain sites along with a higher probability of poor work ability among those with poor team support or a poor possibility to influence (Appendix A). The blue- and white-collar employees also demonstrated a clear difference, which increased with a higher number of pain sites and a higher probability of poor work ability among blue-collar employees (Appendix A).

## 4. Discussion

This study extends previous research which has examined the impacts of the work environment and MSP on work ability over a six-year follow up period. Three different trajectories of work ability were identified over the six years: decreasing, increasing and good work ability. In the former two trajectories, the mean score of work ability changed from good to poor and poor to good during follow-up, while in the latter, individuals maintained good work ability during the follow-up. The number of pain sites experienced by an individual was predictive of being in the pathway of poor work ability.

### 4.1. Identification of Work Ability Pathways

Most employees maintained good work ability over the six years of follow-up, with a small percentage decreasing and increasing their work ability. Consistent with these findings a US-based study also reported three trajectories of work ability with 74% having good work ability, 17% declining and only few, 9% having poor work ability [11]

For the current study, of note is the relative stability of the patterns over the follow up period, suggesting that sustained efforts are required to change the work ability pathway. Interventions designed to target improvements to work ability need to take this into account. A previous exercise-based intervention of 40 weeks duration found no change in work ability, despite other benefits in reducing neck and shoulder pain [31]. A recent systematic review [32] which examined the role of workplace interventions on work ability reported a modest impact. The quality of the evidence base was a contributing factor to this finding; however, the length of follow up for the interventions was also considered an issue. Given the relatively stable nature of work ability, interventions designed to facilitate improvements are likely to take time to see gains and this was not reflected in the time allowed for follow up in studies included in the review.

### 4.2. Predictors of Work Ability Pathways

Pain in more than two body sites was predictive of membership in the poor work ability pathway with the magnitude of association increasing with the number of pain sites recorded. Although the baseline measures of physical and psychosocial working conditions were significant, these did not remain significant once other variables had been controlled for. It is somewhat unexpected that these working conditions were not predictive of work ability but perhaps not surprising given the high proportion of blue workers who are engaged in physically demanding work. One plausible explanation is that MSP is a more proximal measure of work ability than the working conditions. That is, given the previously reported influence of pain on employees needing to leave work early, MSP is more strongly linked with workability than the psychosocial factors as demonstrated by the current results.

Previous research has identified a range of workplace factors associated with work ability, which were not replicated in the current study. Individuals with higher managerial position, high job control and supportive organizational climate were related to the favorable change in work ability among Finnish managers [10]. Similarly, individuals with high mental and physical strain were related to the trajectories of poor work ability in Finnish municipal employees followed from midlife employment until retirement and old age [9].

That no significant association between physical and psychosocial working conditions and work ability were found should not suggest that it is not of importance to identify workplace hazards. Substantial evidence links working conditions with MSP and any improvements may result in subsequent changes in work ability. Work organisations are complex and require systematic approaches to identify and then manage hazards in relation to employee’s health to ensure that all relevant aspects of the environment are considered.

The issue of MSP requires attention, and workplaces need systems in place to monitor musculoskeletal pain levels and implement actions to reduce the hazards associated with the development of pain. A consensus statement developed by the Scientific committee on Musculoskeletal disorders of the International Commission on Occupational health supports this notion, and states: “Musculoskeletal discomfort that is at risk of worsening with work activities, and that affects work ability or quality of life, needs to be identified”, p.3 [33].

Currently, workplaces do not routinely undertake hazard surveillance of workplace factors associated with their employees’ pain and discomfort [34]. A general mistrust of using employee ratings to inform workplace risk management [35] contributes to this and a continued reliance on observational methods despite issues with their validity and reliability [36]. Whilst risk management is not a core focus of the current study, the important role of MSP in determining work ability pathways suggests the need for a greater focus on determining what actions are required to reduce MSP given its important relationship with work ability. Workplace policies and practices need to include mechanisms to ensure that monitoring of all relevant hazards is undertaken on a regular basis.

### 4.3. Strengths and Limitations

A key strength of the current study is the prospective design with six years of follow up. The long follow up provides sufficient time to examine the influences of working conditions and MSP on work ability. The inclusion of blue collar workers who are at higher risk of disability and early retirement in comparison to collar workers is a strength.

A potential limitation is that participants were included in the analysis who may not have responded to all four surveys. Data were analysed for those subjects who replied to work ability questions in all four surveys (*n* = 327) and compared to those who did not respond to all four surveys. The trajectory shapes and group proportions were comparable for both the full and the partial responding groups. Individuals were asked to report musculoskeletal pain in the past seven days, which reduces recall bias but also does not take into account episodic pain which occurs over longer time periods.

The anonymous nature of the data collection did not enable the determination of whether respondents differed from the non-responders with regard to demographics and work ability at study commencement. The healthy worker effect may have an influence here, as those with significant problems may have left the organisation, and the follow up analysis captures those who have remained at the workplace.

Using a median cut off point for the development of the MSP measure may result in some information loss but ensures sufficient cases in each category. To support the development of the measure here, previous studies which have employed this approach were used to guide the process [37,38]. Information on lifestyle factors such as smoking, body mass index and physical exercise was not collected at baseline and not included in the current analyses, although these factors may be related to MSP.

## 5. Conclusions

Findings from this study indicate that multisite pain has an important influence on work ability trajectories. Workplaces addressing the adverse working conditions associated with the development of musculoskeletal pain are likely to reap benefits in the reduction of multisite pain, as well as longer-term improvements in work ability and the likelihood of individuals being able to remain at work.

## Figures and Tables

**Figure 1 ijerph-16-02363-f001:**
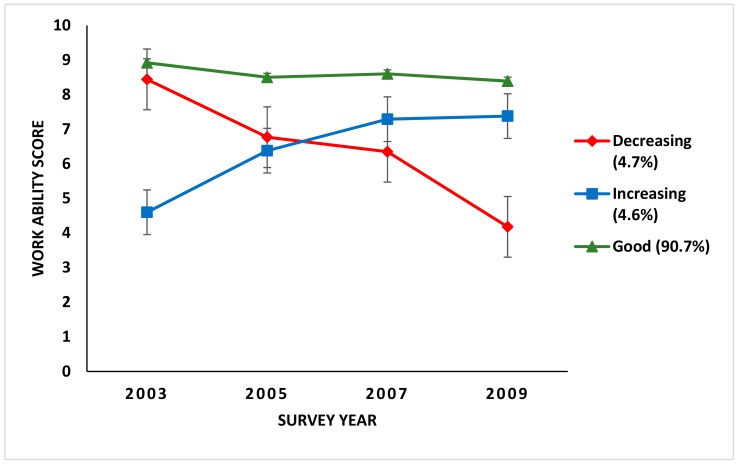
Trajectories of work ability from 2003–2009 in food industrial workers (*N* = 866).

**Table 1 ijerph-16-02363-t001:** Baseline characteristics of the study population according to work ability trajectories.

Baseline Characteristics	Total ^§^ *N* = 866	Work Ability Trajectory (*n*, %)			
		Decreasing (*n* = 41)	Increasing (*n* = 40)	Good (*n* = 786)	*p*-Value ^†^
Age					0.641
<45 years	543	28 (5.2)	26 (4.8)	489 (90.0)	
≥45 years	323	13 (4.0)	13 (4.0)	297 (92.0)	
Gender					0.070
Women	603	32 (5.3)	22 (3.7)	549 (91.0)	
Men	268	9 (3.4)	18 (6.7)	241 (89.9)	
Occupational class					0.003
Blue-collar	601	31 (5.2)	36 (6.0)	534 (88.8)	
White-collar	267	9 (3.4)	4 (1.5)	254 (95.1)	
Physical strain					0.022
Low	374	16 (4.3)	9 (2.4)	349 (93.3)	
High	494	25 (5.0)	31 (6.3)	440 (88.7)	
Repetative movements					0.031
Low	330	9 (2.7)	11 (3.3)	310 (93.9)	
High	539	32 (5.9)	29 (5.4)	478 (88.7)	
Awkward Posture					0.032
Low	353	12 (3.4)	10 (2.8)	331 (93.8)	
High	515	29 (5.6)	30 (5.8)	456 (88.5)	
Mental strain					0.331
Low	410	19 (4.6)	14 (3.4)	377 (92.0)	
High	455	22 (4.8)	25 (5.5)	408 (89.7)	
Leadership					0.199
Good	455	21 (4.7)	15 (3.4)	409 (91.9)	
Poor	403	19 (4.7)	24 (6.0)	360 (89.3)	
Team support					0.005
Good	456	19 (4.2)	11 (2.5)	418 (93.3)	
Poor	407	21 (5.3)	28 (7.0)	349 (87.7)	
Possibility to influence					0.007
Good	457	17 (3.8)	11 (2.5)	418 (93.7)	
Poor	397	21 (5.4)	26 (6.6)	345 (88.0)	
Number of pain sites					0.009
None	233	7 (3.0)	4 (1.7)	222 (95.3)	
One	151	8 (5.3)	5 (3.3)	138 (91.4)	
two	172	5 (2.9)	7 (4.1)	160 (93.0)	
Three	128	7 (5.5)	9 (7.0)	112 (87.5)	
Four	171	13 (7.6)	15 (8.8)	143 (83.6)	

^†^*p*-value derived from the Pearson Chi-Square test; ^§^ The total of each individual variables may not be 100% because of the missing cases.

**Table 2 ijerph-16-02363-t002:** Association of poor work ability pathways with baseline multisite pain from logistic regression models.

Characteristics	OR, 95 % CI for Poor vs. Good Work Ability			
	Model I	Model II	Model III	Model IV
Number of pain sites				
0	1	1	1	1
1	1.90 (0.83–4.36)	1.88 (0.81–4.35)	1.97 (0.83–4.68)	1.96 (0.82–4.65)
2	1.51 (0.65–3.52)	1.40 (0.58–3.38)	1.34 (0.55–3.28)	1.31 (0.52–3.29)
3	2.88 (1.29–6.42)	2.63 (1.13–6.17)	2.52 (1.06–6.00)	2.45 (1.00–6.00)
4	3.95 (1.91–8.19)	3.31 (1.46–7.51)	3.09 (1.36–7.01)	2.96 (1.25–7.03)
Age				
<45 years	1	1	1	1
≥45 years	0.84 (0.51–1.37)	0.86 (0.51–1.44)	0.85 (0.50–1.44)	0.86 (0.50–1.47)
Gender				
Women	1	1	1	1
Men	1.09 (0.66–1.78)	1.35 (0.80–2.27)	1.38 (0.81–2.34)	1.41 (0.82–2.43)
Occupational class				
Blue-collar	1	1	1	1
White-collar	0.42 (0.23–0.77)	0.53 (0.25–1.10)	0.59 (0.29–1.20)	0.62 (0.28–1.37)
Physical strain				
Low	1	1		1
High	1.74 (1.06–2.85)	0.99 (0.54–1.80)		1.05 (0.57–1.96)
Repetitive movements				
Low	1	1		1
High	1.94 (1.15–3.29)	1.27 (0.64–2.51)		1.24 (0.60–2.58)
Awkward Posture				
Low	1	1		1
High	1.90 (1.14–3.17)	0.94 (0.46–1.91)		0.91 (0.43–1.93)
Mental strain				
Low	1		1	1
High	1.28 (0.80–2.05)		1.04 (0.62–1.74)	1.03 (0.61–1.74)
Leadership				
Good	1		1	1
Poor	1.36 (0.85–2.16)		0.93 (0.55–1.57)	0.93 (0.55–1.57)
Team support				
Good	1		1	1
Poor	1.96 (1.22–3.15)		1.57 (0.92–2.68)	1.56 (0.91–2.66)
Possibility to influence				
Good	1		1	1
Poor	2.03 (1.25–3.31)		1.31 (0.74–2.31)	1.27 (0.71–2.27)

Model I: Crude model; Model II: Adjusted for age, gender, occupational class and the physical factors at work.; Model III: Adjusted for age, gender, occupational class and the psychosocial factors at work; Model IV: Simultaneously adjusted for all variables included in Model I.

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
