# Peer review of "What Are the Key Workplace Influences on Pathways of Work Ability? A Six-Year Follow Up"

_ijerph, 2019, doi:10.3390/ijerph16132363_

Round 1

Reviewer 1 Report

Recommend publication after the following items in lines are corrected.

The abstract should be written to encourage readers rather than showing specific results. Maybe better to leave results out for further investigation of the article by readers.

Line 19 - the word later should be latter

Lines 58, 130, 131, 237 - Should not begin sentence with an acronym.

Line 63 - Instead of using aim of study, write Purpose of Study with a separate heading (1.1).

Line 69 -Capitalize Food

line 70 - No need for hyphens

Line 71 - latter is correct usage

Line 76 - The symbol % is for charts and graphs; write percent when presented in text format

Line 137 text should also be just before the figure as an introduction to the figure.

Line 163 Paragraph should follow Figure 1.  Basically, there should be an introduction to a figure (or table) just before it, and a summary after it.

Lines - 170, 178, 216, 222, 225, 237, 279 - Single digit numbers should be written - e.g., nine.

Line 263 - 264.  Direct quotes should have specific page numbers in references.

Reference journal title words should be capitalized.  You have some that are and some that  are not.

Reviewer 2 Report

Dear authors, 

The time involved in submitting your manuscript is greatly appreciated.

Despite this, the article presents a series of issues that must be noted and mended. The recommendations are presented separately by sections. Hopefully, they would be useful. Firstly, some of the references that you cite are too old. Even though the most relevant studies should be referenced, also the RECENT research must be included. Moreover, I recommend a strong effort in explaining the framework of Workability as the theoretical umbrella that covers your research. Moreover, about all the paper, I would recommend avoiding a lot of abbreviations. It is too hard for readers. Due that the Journal does not have any limitation in the number of words, please, try to reduce these abbreviations.

Secondly, at the end of the literature review, the aims and the questions in the research should appear. Maybe to formulate the questions as hypotheses would be an option to clear this aspect. Another commentary, it is the possibility of including this part at the final of the introduction part; even a separate section could be a good option, in order to clear the final of the introduction and to serve as a connection with the method. Moreover, between lines 63 and 66 the text seems copied and pasted from another source. Please, check it. 

Thirdly, related to the Method:

Please, try to better describe the sociodemographic data of your participants. In the same sense, give the readers with detailed information about the procedure for recruiting participants and collecting data.

Fourthly, related to the instruments: please better inform about their psychometric quality and give to the readers some example of the items. If you can, please inform about previous studies where the same instrument has been used and the reliability obtained in that research.

 I think that the results section is very good, despite the use of abbreviations, that difficult the task for readers.

Finally, a section related to future lines of investigations and the principal contributions of the research could be interesting. Suggestions for future research should be based not only on the limitations of the present research. Your paper has a lot of relevant implications for Human resource managment, but also for society and policymakers, but you need to elaborate more on this topic.

Reviewer 3 Report

The manuscript provides confirmatory evidence to support the current understanding, it does not bring forward any particular novelty in terms of method, originality of the approach, or non-conformity of the results. English should be improved throughout the manuscript.
